# Methods for estimating the burden of acute tropical infectious diseases: A scoping review

Qian Wang[1,2]*, Richard James Maude[1,2,3], Nicholas Philip John Day[1,2], Benn Sartorius[2,4,5]

**1** Mahidol Oxford Tropical Medicine Research Unit (MORU), Faculty of Tropical Medicine, Mahidol University, Bangkok, Thailand, **2** Centre for Tropical Medicine and Global Health, Nuffield Department of Medicine, University of Oxford, Oxford, United Kingdom, **3** The Open University, Milton Keynes, United Kingdom, **4** Centre for Clinical Research (UQCCR), Faculty of Health, Medicine and Behavioural Sciences, University of Queensland, Brisbane, Australia, **5** Department of Health Metric Sciences, School of Medicine, University of Washington, Seattle, Washington, United States of America

* Qian@tropmedres.ac

## Abstract

Acute infectious diseases, particularly lots of neglected tropical diseases (NTDs), pose significant public health challenges, especially in resource-limited settings where diagnostic and surveillance capacities are often inadequate. This scoping review systematically explores methodologies for estimating the burden of acute infectious NTDs, focusing on metrics such as incidence, mortality, and disability-adjusted life years (DALYs). We identified 60 studies, predominantly on malaria and dengue, with a growing emphasis on advanced computational approaches like machine learning and Bayesian geospatial modeling. Key findings highlight the evolution from traditional surveillance-based methods to integrated frameworks incorporating environmental, demographic, and health system covariates. However, challenges persist, including data sparsity, underreporting, and methodological uncertainties. The review underscores the need for improved data integration, standardized frameworks, and interdisciplinary collaboration to enhance the accuracy and utility of burden estimates.

## Author summary

We conducted this study because many short-term infectious diseases, especially those neglected tropical diseases affecting people in tropical and low-income areas, are often overlooked in public health planning. These diseases can spread quickly and cause serious illness or even death, but reliable data on how many people are affected is often missing. Our goal was to understand how scientists estimate the burden of these diseases, especially when information is limited. We reviewed 60 published studies that used different methods, from basic counting of reported cases to complex models that use geography, climate, and population data to fill in the gaps. We found that diseases like malaria and

**Data availability statement:** This study did not generate or analyse primary data. All data used are from published sources, which are cited in the manuscript. The full list of included studies is provided in the Supporting information.

**Funding:** This research was funded in part by the Wellcome Trust [220211] to QW, RJM and ND. BS was supported by an Australian National Health and Medical Research Council Investigator Grant (GNT2034827) and the Operational Research and Decision Support for Infectious Diseases (ODeSI) program, funded by The University of Queensland's Health Research Accelerator (HERA) initiative (2021–2028). The funder had no role in study design, data collection, data analysis, data interpretation, or writing of this study.

**Competing interests:** The authors have declared that no competing interests exist.

dengue are more commonly studied, while other diseases receive much less attention. Newer methods, such as machine learning and mapping tools, offer powerful ways to improve burden estimates. However, challenges like missing data and inconsistent reporting remain. By summarizing what has been done so far, our work provides a roadmap for future studies to better estimate the impact of these diseases and help governments and health organizations make more informed decisions.

## Introduction

Acute infectious diseases are a major public health challenge worldwide, particularly in tropical and resource-limited settings, where they contribute significantly to morbidity, mortality, and healthcare resource utilization [1]. These diseases, often characterized by their rapid onset and potential for widespread outbreaks, encompass a broad spectrum of conditions, including viral, bacterial, parasitic, and fungal infections [2]. Many neglected tropical diseases (NTDs) and other neglected infectious diseases form a critical subset of acute infectious diseases that disproportionately affect vulnerable populations in resource-limited settings, and where diagnostic and surveillance capacities are often inadequate [3]. Many of these diseases, such as dengue, leptospirosis, scrub typhus, and chikungunya, share nonspecific clinical presentations, complicating accurate diagnosis and surveillance [4,5]. Limited access to diagnostic facilities in low-resource settings often forces reliance on empirical treatment, leading to underreporting and misclassification. For instance, diseases like dengue and leptospirosis are frequently underdiagnosed [6,7], while others, such as scrub typhus and chikungunya, remain overlooked in endemic regions [8,9]. Emerging infections such as Zika virus also lack robust surveillance systems [10], leaving significant gaps in data collection and burden estimation. Despite their significant health and socioeconomic impact, these conditions frequently remain underreported and inadequately addressed, in part due to their systematic de-prioritization in global health funding. This issue has been exacerbated by recent shifts in foreign policy and funding reallocations, such as the downsizing of United States Agency for International Development (USAID) [11] and other major funding mechanisms that traditionally supported infectious disease surveillance and control in low-income settings. Consequently, the true burden of these diseases is not only underestimated but, in some cases, completely lacking, undermining efforts to inform targeted interventions and resource allocation in the most affected regions.

Malaria and dengue illustrate these challenges particularly clearly. Malaria remains a leading cause of morbidity and mortality in sub-Saharan Africa, with hundreds of millions of cases annually and a disproportionate burden among young children [12,13]. Although geospatial and modelling approaches have improved incidence and mortality estimates [13,14], uncertainty remains high in settings with weak surveillance systems and heterogeneous treatment-seeking behaviour, complicating the targeting and evaluation of control interventions [12,15]. Similarly,

dengue has expanded rapidly across tropical and subtropical regions, and model-based analyses consistently show that reported case counts substantially underestimate the true burden due to under-ascertainment, misdiagnosis, and a high proportion of asymptomatic infections [16–19]. These persistent gaps between reported and estimated burden hinder optimal resource allocation, vaccine deployment strategies, and long-term elimination planning, underscoring the need for robust and transparent methodological frameworks for burden estimation.

Robust and transparent methods for estimating disease burden, using metrics such as incidence, mortality, and disability-adjusted life years (DALYs), are therefore essential for informing policy and prioritization. The Global Burden of Disease (GBD) study, initiated in the early 1990s, established a standardized and comparative framework for quantifying health loss across diseases, injuries, and risk factors worldwide [20–23]. Central to this framework is DALY, a composite metric that combines years of life lost due to premature mortality (YLL) and years lived with disability (YLD). The World Health Organization (WHO) adopted and further refined this DALY-based methodology, providing standardized approaches for calculating YLL based on life expectancy tables and YLD based on disability weights and disease duration. These frameworks have enabled cross-disease and cross-country comparisons and have become foundational tools in global health priority setting. While these frameworks have enabled global benchmarking, many acute tropical infectious diseases present distinct methodological challenges, including dynamic transmission patterns, spatial heterogeneity, incomplete surveillance, and substantial underreporting. In response, a wide range of approaches, from traditional epidemiological models to machine learning and Bayesian geospatial methods, have been developed to address these complexities [24,25]. However, the strengths, limitations, and evolution of these diverse methodological frameworks have not been systematically synthesized.

This scoping review therefore aims to address this need by systematically identifying, exploring, and summarizing the literature on methods for estimating the burden of acute infection diseases, with particular attention to NTDs. By examining the volume, nature, and characteristics of primary research, we seek to identify gaps, highlight methodological innovations, and provide clear guidance for advancing burden estimation approaches, ultimately guiding more robust burden estimation practices for underrecognized diseases.

## Method

### Study design

This scoping review was conducted to identify, explore, and summarize the literature on methods for estimating the burden of acute infectious diseases. This scoping review was guided by the Population–Concept–Context (PCC) framework for scoping reviews [19]. The Population comprised populations affected by acute tropical infectious diseases. The Concept focused on methodological approaches used to estimate disease burden, including models, analytical frameworks, and burden metrics (e.g., incidence, mortality, DALYs). The PCC framework informed the development of the research questions, eligibility criteria, and data charting process. The Context included all geographic regions and settings where such burden estimation studies were conducted. The methodology was structured to address the following research questions: (1) What burden estimation methods for acute infectious diseases have been used in the literature? (2) How have these methods evolved over time? (3) What are the key factors influencing the selection of burden estimation methods? (4) What are the strengths and limitations of different methods? (5) What are the gaps and opportunities for further research? A detailed protocol has been registered on the Open Science Framework (Registration https://doi.org/10.17605/OSF.IO/FSZ5B).

### Search strategy and study selection

The search strategy used a combination of search strings related to outcomes (disease burden, DALYs, etc.), methods (method, model, approach) and diseases (infectious disease, communicable disease, or infection). The full electronic

search strategy is provided in S2 Table. Searches were conducted in PubMed, which provides comprehensive indexing of biomedical, epidemiological, and global health literature, including the core journals in which infectious disease burden modelling studies are typically published, to identify relevant studies covering publications from 1990 to 2025. The database search was restricted to publications from 1990 onwards; however, no publication year restrictions were applied to studies identified through reference screening. The original search was done in 30 March 2024 and updated in October 2025. One reviewer (QW) conducted title/abstract and full-text screening using predefined inclusion and exclusion criteria. A second reviewer (RJM) independently reviewed uncertain or ambiguous cases, and disagreements were resolved through discussion to ensure consistency in study selection. Additionally, we used a 'snowballing' method [26], conducting further screening of references cited in the included articles to identify and incorporate additional relevant records into our final list. The identified records were imported into Rayyan, a web-based systematic review management tool that enables blinded screening, reviewer conflict resolution, and machine learning–assisted prioritization to streamline study selection [27].

We included studies that focused on acute tropical infectious diseases within NTDs list NTDs recognized by PLOS Neglected Tropical Diseases [28] and more neglected diseases as scrub typhus, employed quantitative or qualitative methods to estimate disease burden, were published in peer-reviewed journals in English, and addressed any geographical region or population. Articles were included in the review if they met the following criteria: (1) the article employed a model or framework to estimate the health burden (morbidity, mortality, DALYs, etc.) of one or more acute infectious diseases; (2) the article was published in a peer-reviewed journal; and (3) the article was available in English. Certain study types were explicitly excluded to maintain methodological clarity and focus on innovative modeling approaches. Excluded studies included those: (1) exclusively based on long-term observational epidemiological cohorts or surveillance data to directly estimate incidence without applying any additional modeling or analytical framework; (2) that directly applied the World Health Organization (WHO) DALYs calculation methods without further methodological advancement or modification; and (3) that solely extracted, analyzed, or presented disease burden estimates directly from the Global Burden of Disease (GBD) database without novel methodological contributions. Studies were excluded if they lacked methodological details relevant to burden estimation or focused solely on chronic infectious diseases or non-infectious conditions. Specific exclusion criteria included: (1) the article did not focus on acute tropical infectious diseases; (2) the article didn't mention any specific methods; (3) the article only measured economic cost or social burden; or (4) the full text of the article could not be obtained; or the article was not in English.

## Data charting

For each included article, data were extracted and charted in a standardized form regarding: (1) general information about the articles, e.g., authors, titles, publication year; (2) aim(s) of the research; (3) geographical regions where the research was conducted; (4) disease(s) studied; (5) method used; (6) base data and data source; (7) covariates included in the model; (8) metrics of disease burden; (9) outputs and key findings; (10) limitations; (11) techniques to address uncertainty in the burden estimates. We did not appraise methodological quality or risk of bias of the included articles, which is consistent with scoping review guidance [29].

Our goal was not to obtain numerical findings from the individual studies, but rather to summarize the main analytical strategies and data used to estimate the disease burden of acute infectious diseases. We conducted a descriptive analysis to summarize the distribution of methods across diseases, regions, and time periods. The evolution of methods over time was examined, and the limitations of commonly used techniques were qualitatively assessed. To provide actionable insights, the review also explored how different modeling approaches addressed spatial resolution, uncertainty quantification, and the integration of covariates.

This scoping review adhered to PRISMA-ScR (Preferred Reporting Items for Systematic Reviews and Meta-Analyses extension for Scoping Review) guidelines [30] (S1 Table).

## Ethical Considerations

As this study involved a review of publicly available literature, no ethical approval was required.

## Result

### Basic characteristics

A total of 2,181 records were identified via the search strategies of PubMed and loaded into Rayyan. After the screening of titles and abstracts, 283 articles were retained for the full text screening, with 1,898 records removed because of irrelevant topics. A total of 41 studies were included in this review after full-text screening from the PubMed database. In addition, 89 records were identified and assessed through reference screening ("snowballing"), of which 19 met the inclusion criteria. Overall, 60 studies were included in the final review. Figure 1 depicts the search and screening process following the Preferred Reporting Items for Systematic reviews and Meta-Analyses (PRISMA) guidelines [30,31]. The full list and basic characteristics of all the included studies can be found in appendix (S3, S4 Tables).

The majority of studies (n = 47, 78%) were conducted between 2010 and 2022, while 17% were published between 2000 and 2010, and 5% before 2000 (S6 Table). Regarding the geographic focus, the African Region (AFRO) accounted for the highest proportion of studies (n = 24, 40%), followed by global studies (n = 16, 27%), and South-East Asian Regions (SEARO) (n = 12, 20%) (Fig 2). The Western Pacific (WPRO) and Region of the Americas (AMRO) both represented 15% and there were 1 from the Eastern Mediterranean (EMRO). In terms of diseases studied, malaria and dengue were the most frequently investigated, accounting for 21 (35%) and 22 (37%) of included studies, respectively. Other diseases examined included yellow fever (13%), leptospirosis (5%), scrub typhus (3%), and Japanese encephalitis (3%). The

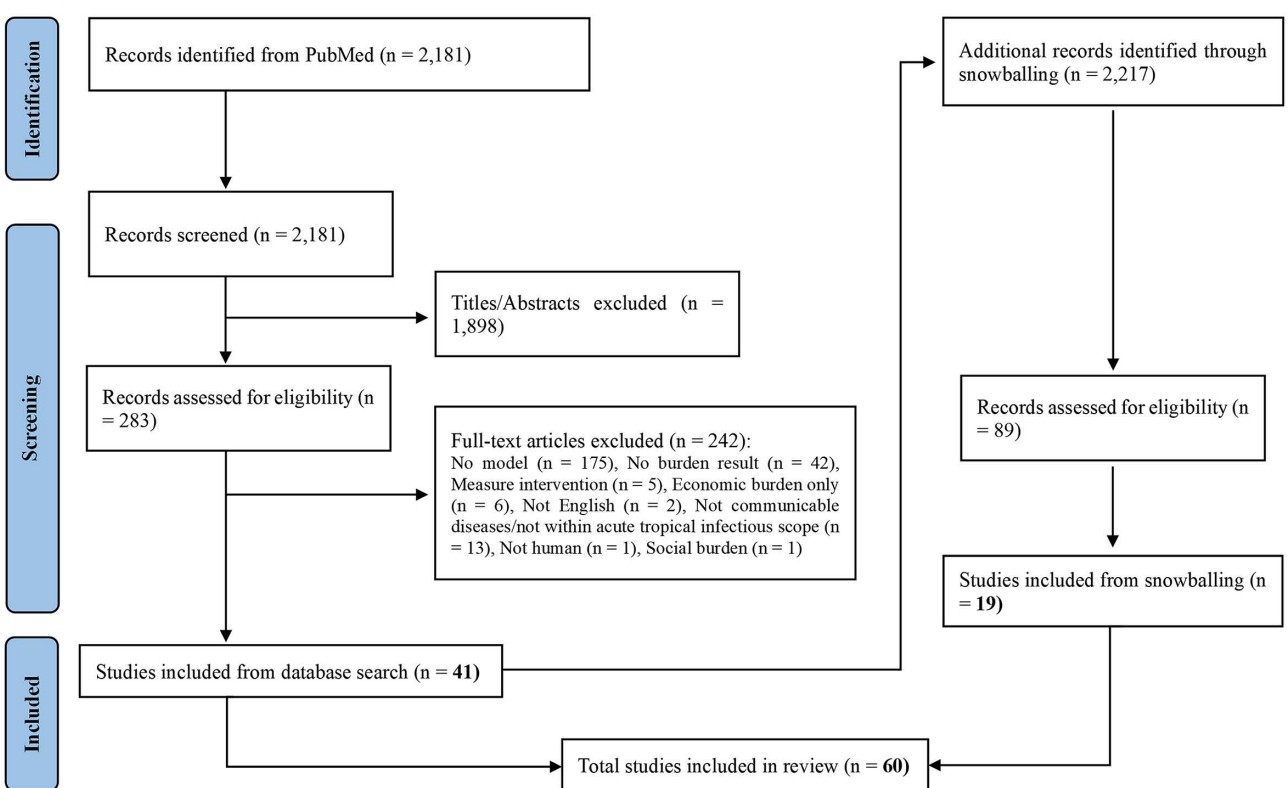

**Fig 1. Flow chart outlining the search and review process, the records identified, included, and excluded as well as the reasons for exclusion.**

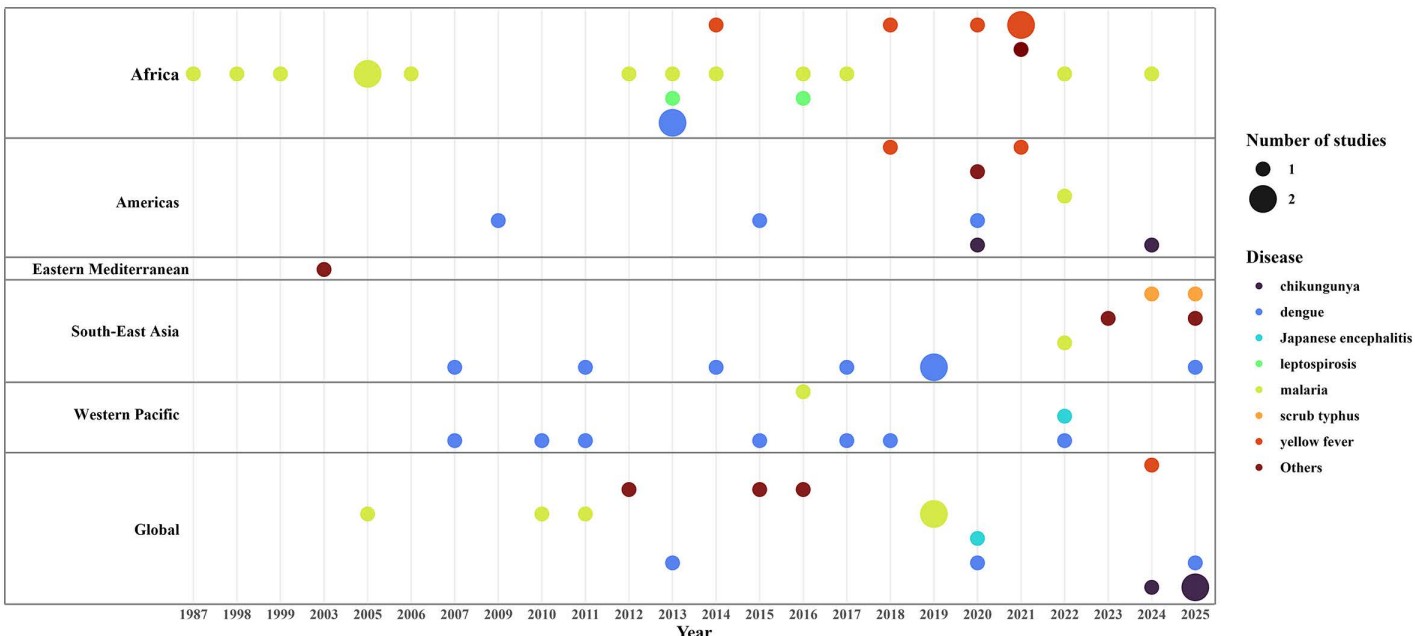

**Fig 2. Publication year and geographic distribution of included studies (n = 60).** Bubble position shows year and WHO region; bubble color indicates disease (low-frequency (total = 1) diseases combined as "Others"); bubble size indicates number of studies (1 or 2).

remaining 2% each (one study per disease) comprised Zika virus, typhoid fever, leishmaniasis, spotted fever group rickettsioses, murine typhus, melioidosis, schistosomiasis, and Q fever (Fig 2).

## Base data, sources, spatial resolution and extent

Estimates of the disease burden were primarily derived by combining various sources, including routine surveillance systems (which capture incidence data), prevalence surveys (which provided prevalence data), outbreak reports, cohort studies, and other structured epidemiological studies. Online reporting platforms, which always are part of formal surveillance systems or function, were also considered. Additionally, clinical case records, including physician-reported cases and hospital-based diagnoses, contributed to burden estimates, particularly in settings with limited formal surveillance. The choice of data sources and methodologies for estimating the health burden of infectious diseases depended on the specific disease's epidemiological characteristics, the availability and quality of data, as well as the surveillance and diagnostic capacities in the affected regions. The spatial extent and spatial resolution also varied with different diseases. High-resolution spatial data at point or pixel levels are preferred for diseases with more granular surveillance and reporting systems, while coarser resolutions with data aggregated at administrative unit level (e.g., province or country) are often used in resource-limited settings.

Malaria burden estimation relies on diverse base data, including empirical data input and previously modelled outputs [12–14,32–46]. Empirical base inputs include reported cases from national surveillance systems, parasite prevalence surveys at subnational or community levels, incidence and mortality data derived from household surveys. These empirical inputs are sourced from large-scale initiatives such as Demographic and Health Surveys (DHS), Multiple Indicator Cluster Surveys (MICS) reports [44], WHO reports [12], and country-specific surveillance databases(e.g., Cambodia Malaria Database) [43]. In addition to these primary data sources, some studies incorporate previously modelled outputs, such as geospatial risk or prevalence maps produced by initiatives like the Malaria Atlas Project (MAP) [14,46], as secondary input

layers. While these model-derived surfaces are not raw base data, they are sometimes used as covariate inputs to inform subsequent burden calculations. Additionally, literature reviews and subnational surveys are frequently used to fill data gaps, particularly in resource-limited settings [40]. Supported by relatively rich data sources, many of the malaria studies focus on global [13,37,39,45] or multi-country spatial extents [14,42], leveraging high-resolution data at the point level (e.g., specific survey locations) [13,32,34,37,39,45] or aggregated at administrative boundaries (e.g., districts or provinces) [33,44], which allow for detailed mapping of malaria prevalence and incidence across diverse settings.

Dengue burden estimation similarly relies on a variety of base data, including reported cases from national surveillance systems, dengue seroprevalence data from surveys, incidence and mortality data, as well as age-stratified incidence data derived from cohort studies and literature reviews [16–18,47–59]. These data are sourced from a combination of literature passive surveillance systems (e.g., routine case reporting by healthcare facilities), and active surveillance efforts (e.g., targeted outbreak investigations, sentinel surveillance sites, and serological surveys conducted by government ministries or international organizations such as WHO [49], and expert opinions, which help address gaps in routine surveillance data [52]. Compared to malaria, dengue studies mainly focus on country-level spatial extents [18,57,59], although some analyses extend to multi-country regions [17,49,53] or highly specific contexts, such as individual schools [47]. Spatial resolutions range from administrative boundaries (e.g., city, state, or country levels) to fine-grained grid-based resolutions (e.g., 5×5 km grids) [18,57,58]. Importantly, these high-resolution grid estimates represent modelled outputs which are used to capture detailed geographic variations in disease burden rather than primary base data, although they may subsequently be used as inputs for downstream burden calculations.

For other acute infectious diseases, the burden estimation literature is relatively sparse and also relies on diverse base data and sources tailored to the specific epidemiology and surveillance capabilities of each disease. For example, yellow fever studies use occurrence data, serological surveys, and case reports, sourced from literature, online databases, and historical outbreak records, with spatial resolutions ranging from province-level to pixel-level grids for multi-country and global extents [60–63]. Chikungunya burden estimation relies on reported case data from ministries of health, regional platforms, seroprevalence studies and other published literature, typically analyzed at subnational to country-level resolution for national or global estimates [64–68]. Scrub typhus burden estimation relied on longitudinal antibody data from cohort and population serosurveys as well as sentinel hospital surveillance surveys, analyzed at study-site or household level resolution, with spatial extents ranging from community to state level [69,70]. Japanese encephalitis virus (JEV) burden estimation leverages age-stratified case data from literature and national reports, typically analyzed at the country level across multiple regions [59,71]. Similarly, leptospirosis studies utilize published morbidity and mortality data, as well as population and fever surveillance data from hospital-based systems, focusing on country-level resolutions [72,73]. For leishmaniasis, burden estimation incorporates expansion factors derived from published and official reports, combined with expert opinion, with a spatial resolution of country-level to provide an extent of global estimates [74]. For melioidosis highly detailed datasets were used, including human and animal case reports, analyzed at a pixel-level resolution to produce global estimates of disease burden [75].

## Burden metrics

The common metrics to measure the burden of disease are case/infection numbers (31 of 60) and incidence (19 of 60), deaths (16 of 60) mortality (8 of 60) and DALYs (12 of 60), see Table 1. DALYs were estimated in 11 studies, including six focused on dengue, one on malaria and four on chikungunya (Table 2). Although malaria incidence and mortality have been extensively studied, relatively few included studies estimated malaria DALYs using novel modelling approaches, likely reflecting the exclusion of studies that directly applied standardized WHO or GBD DALY calculation frameworks without methodological modification. Some studies used novel metrics like attributable fever days [25] and severe admission rate to measure the clinical burden of malaria [44]. Transmission intensity/force of infection [57,58,60,63,71], prevalence [13,37,40,45,63], occurrence risk [5,37,61,63,75], and endemicity [2,33,39] have been utilized as intermediate

**Table 1. Summary of burden metrics used in disease burden estimation, stratified by disease type and region.**

| Burden Metric | No. Studies | Examples of Diseases | Regions with Studies |
|---|---|---|---|
| Case/infection numbers | 31 | malaria [12,33,37,39,40,42,45,76], dengue [16–18,49–50,52,54,55,57,58,77,78], leishmaniasis [74], yellow fever [60,61,63], chikungunya [66–68], Japanese encephalitis [71], schistosomiasis [79] | Africa, Americas, South-East Asia, Western Pacific, Global |
| Incidence | 19 | Malaria [12,13,35,42,45,46], leptospirosis [72,73,80], melioidosis [75], dengue [18,56,59], scrub typhus [69,70], Japanese encephalitis [59], Q fever and spotted fever group [81], chikungunya [66], typhoid fever [82] | Africa, Americas, South-East Asia, Western Pacific, Global |
| Death number | 16 | malaria [33,36,38,76], dengue [49,77,83], yellow fever [60,63], chikungunya [66,67], Japanese encephalitis [71], leishmaniasis [74] | Africa, Americas, South-East Asia, Western Pacific, Global |
| Mortality rates | 8 | malaria [13,14,32,34,44], leptospirosis [73], melioidosis [75], dengue, chikungunya and Zika [64] | Africa, Americas, Global |
| Disability-Adjusted Life Years (DALYs)/YLD | 12 | dengue [47,48,49,51,55,57,84], malaria [76], chikungunya [64–67] | Africa, Americas, South-East Asia, Western Pacific, Global |
| Attributable fever days | 1 | malaria [41] | Africa |
| Severe admission rates | 1 | malaria [44] | Africa |

**Table 2. Summary of covariates used in disease burden estimation.**

| Covariate Type | Examples | Relevant Diseases | Use in Burden Estimation |
|---|---|---|---|
| Environmental | Temperature (mean, range, max, min), Precipitation (annual, seasonal), NDVI, EVI, Land Cover, Altitude, Soil Type, Vegetation/Moisture Indices | Malaria, Dengue, Yellow Fever, Japanese Encephalitis, Melioidosis | Defines transmission limits and risk zone, enables spatial prediction and extrapolation in geospatial models |
| Demographic | Population Size, Age Stratification, Urban vs. Rural, Population Density | Pediatric Malaria, Dengue | Refines incidence estimates, age-specific burden and population-adjusted burden estimates |
| Social & Economic | Education Level, Healthcare Access, Household Density | Malaria, Dengue, Leptospirosis | Adjusts for healthcare disparities and reporting biases |
| Healthcare-Related | Treatment-Seeking Behavior, Hospitalization Rates, Healthcare System Quality | Malaria, Dengue, Leishmaniasis, Q fever and spotted fever group | Evaluates healthcare access, adjusts observed data, corrects under-ascertainment and severity bias |
| Intervention-Based | Drug Procurement, Vector Control Programs, Coverage of Vaccine | Malaria, Dengue, Yellow Fever | Accounts for the impact of control measures on disease burden, modifies transmission intensity in dynamic models |

metrics to generate the final burden result, and basic reproduction number ($R_0$) used in mathematical models as an internal metric for the final burden result [53].

## Model/method used

A wide range of models has been employed to estimate the burden of acute infectious NTDs. In the late 1990s fuzzy logic models [33,34] and climate suitability frameworks were widely used to explore environmental drivers and stable transmission zones but have since been largely replaced by more advanced approaches. Modern methods, including machine learning algorithms [57–59,75,78] (e.g., random forest, gradient boosted models, neural networks) and Bayesian framework

with geostatistical models and Markov Chain Monte Carlo (MCMC) simulations [13,18,39,40,42,43,45,62,63,71], now dominate the field, enabling the integration of diverse covariates such as environmental, demographic, and socioeconomic factors to refine burden estimates and to handle uncertainty and spatial dependencies. Classical statistical approaches, such as generalized linear models (GLM), linear regression, and Poisson regression, are still employed and remain foundational for many studies, often combined with modern methods to enhance the robustness [43,61,63].

A critical methodological innovation is the ability to infer incidence from prevalence data, specifically through fitting age-specific seroprevalence data to estimate the force of infection in the absence of direct longitudinal incidence data [18,67,71].

Morbidity estimation typically involves adjusting reported case numbers using expansion factors derived from comparisons between active investigations (e.g., active surveillance studies, seroprevalence surveys, or community-based investigations) and official passive surveillance data, often supplemented by expert opinion [48,54,55]. These have been used for dengue [17,49–52,77], leptospirosis [72], leishmaniasis [74], scrub typhus, murine typhus and spotted fever group rickettsioses [70], Q fever and spotted fever group [81]. Mortality estimation frequently relies on applying case-fatality ratios (CFRs) to estimated incidence.

These morbidity and mortality estimates feed directly into disability-adjusted life years (DALYs) calculations. DALYs combine two distinct metrics: Years of Life Lost (YLL), calculated by multiplying deaths at each age by the remaining life expectancy for that age, and Years Lived with Disability (YLD), derived by multiplying incident cases by disability weights and the average duration of illness [85,86]. All those epidemiological models and statistical analyses mentioned above typically provide essential inputs for these calculations, such as age-specific incidence rates, disease durations, and mortality estimates.

Specialized models such as incidence-prevalence-mortality (IPM) [56], simulation-based models [44], and microsimulation approaches [14,32] allow detailed integration of morbidity, mortality, and intervention impacts, thereby providing comprehensive estimates of overall disease burden. Notably, these modeling methods are broadly applicable across different transmission modalities, with specific covariate selections reflecting distinct ecological and epidemiological contexts, detailed below in covariates part.

## Method development over time

The development of methods for estimating disease burden has evolved significantly over time, reflecting advancements in data availability, computational modelling, and surveillance capacity (S1 Fig). This overall trend is particularly well illustrated by the extensive literature on malaria and dengue. For malaria, early approaches relied on manual adjustments to reported case data and the use of risk maps derived from limited surveys. Over time, these methods progressed to include spatially explicit models like the MAP [14], which integrated geostatistical approaches and machine learning models to combine incidence and prevalence data with environmental and demographic covariates. Recent efforts have focused on leveraging high-resolution spatial data (e.g., pixel-level grids) and incorporating climate and geographic features to refine burden estimates at global and multi-country scales [13,45].

Similarly, for dengue, initial estimates relied heavily on reported cases and hospital-based surveillance [48], often underestimating true incidence due to underreporting and misdiagnosis. Over time, methodological advancements introduced expansion factors to adjust reported data [50,54,55], along with the integration of seroprevalence studies, active surveillance, and geospatial models [56,59]. Modern methods now utilize Bayesian frameworks, machine learning algorithms, and dynamic transmission models to account for spatiotemporal variations in incidence, incorporating environmental and socioeconomic factors [18,57,58].

For other acute tropical infectious diseases, methodological development shows a similar shift from simple correction-based approaches to more advanced statistical and transmission-dynamic frameworks. Early estimates for typhoid fever, leptospirosis and leishmaniasis largely relied on multiplier and expansion-factor methods to adjust surveillance data

for under-ascertainment, incorporating provider-sampling adjustments and diagnostic sensitivity corrections [82,74,87]. Regression-based approaches, including generalized linear models and negative binomial models, were later introduced to better account for reporting heterogeneity and covariate effects [88–90]. For environmentally mediated infections such as melioidosis and schistosomiasis, boosted regression trees and Bayesian geostatistical models enabled spatially explicit risk mapping [79,91]. Yellow fever modelling has evolved from GLM-based calibration models to Bayesian hierarchical and mechanistic temperature-suitability frameworks, including serology-informed force-of-infection estimation and ensemble projections developed by collaborative modelling groups [92,93]. Likewise, Japanese encephalitis and chikungunya studies increasingly apply catalytic and sero-catalytic models within Bayesian frameworks, machine learning algorithms, and dynamic compartmental transmission models, often combined with Monte Carlo simulation and DALY-based metrics incorporating underreporting adjustments [66,94].

A significant methodological milestone, relevant across multiple diseases, has been the development and refinement of the Disability-Adjusted Life Years (DALYs) framework. First conceptualized in the Global Burden of Disease (GBD) study by Murray and Lopez in the early 1990s and subsequently adopted by the World Health Organization [85,63], DALYs provide standardized metrics that combine morbidity and mortality into a single unified measure. This has allowed for comparative assessments of disease burden across regions, populations, and different health conditions. Due to its methodological robustness and comparative value, the DALY framework has become widely used in public health research, health impact assessments, and global disease burden estimations [95–97]. Recent studies have advanced DALY computation by incorporating country-specific underreporting adjustments and probabilistic uncertainty propagation using Monte Carlo simulation [65], which has strengthened the comparability and policy relevance of burden estimates while explicitly accounting for structural and data-driven uncertainty.

## Covariates

The studies reviewed demonstrate a wide variety of covariates used in burden estimation across diseases, reflecting diverse transmission modalities, epidemiological contexts, and population vulnerability and health system resilience. These covariates spanned environmental conditions, vector ecology, demographic characteristics, and healthcare system capacity (Table 2).

Environmental variables such as temperature (mean, range, maximum, minimum), precipitation (annual, seasonal), land cover classifications (e.g., forest, urban areas, cropland), enhanced vegetation index (EVI), normalized difference vegetation index (NDVI), and altitude have been widely used for most of those NTDs to capture the climatic and ecological conditions influencing transmission. Particularly for mosquito-borne diseases like malaria [12,36,43,46], dengue [57], yellow fever [61–63], chikungunya [64–67] and Japanese encephalitis [59], climate suitability, temperature suitability, habitat suitability and the mosquito distribution have been generated [98] and included in the burden estimation framework as an important index for defining the potential risk zone, given the vector-specific ecological preferences [99]. Diseases with significant environmental components, such as melioidosis, emphasize soil characteristics, moisture indices, vegetation, and land-use patterns, reflecting their environmental reservoirs and exposure pathways [75].

Demographic factors, including population size, age stratification, urban versus rural classification, and population density, were critical for refining estimates, with age-specific data especially important for diseases like pediatric malaria and dengue. Social characteristics like education levels, healthcare access, and household density were also employed [13,45]. Healthcare-related covariates, such as treatment-seeking behavior [12,39,47,52,74], hospitalization rates [49,51], healthcare system quality and disability weights [32,47,55,72,75] were incorporated to evaluate healthcare access and quantify disease burden. Some interventions like drug procurement and vector control programs were also integrated to adjust estimates [39,42,74].

Importantly, the selection and complexity of covariates reflect both disease ecology and the evolution of burden estimation methodologies. Earlier approaches, particularly those relying on expansion factors or multiplier adjustments to

surveillance data, typically incorporated a limited set of demographic and healthcare-access variables to correct for under-reporting and care-seeking behaviour [82,74,87]. In contrast, more recent geospatial and machine learning-based models integrate high-dimensional environmental and vector ecology covariates to enable spatial prediction beyond locations with empirical observations. For vector-borne diseases such as malaria, dengue, and yellow fever, environmental and climate suitability variables (e.g., temperature, precipitation, vegetation indices, and vector distribution metrics) are central, as they define transmission potential and support fine-resolution spatial interpolation. Meanwhile, healthcare utilization and demographic covariates remain particularly important in approaches aimed at adjusting reported case data. This pattern highlights a broader methodological transition from adjustment-based estimation frameworks toward predictive modelling approaches capable of generating spatially continuous burden surfaces, especially in data-sparse regions.

## Uncertainty

The studies employed various robust statistical and computational techniques to quantify and address uncertainty in burden estimates. Monte Carlo simulations were one of the most used methods, with iterations ranging from 1,000–250,000, allowing for simultaneous variation of input parameters [17,39,40,42,43,49,51,52,55]. These simulations generated plausible distributions for estimates, often presented as mean values with 95% confidence or credibility intervals (CIs). Probabilistic sensitivity analysis, coupled with triangular or beta distributions, was also widely applied to explore parameter uncertainty and generate credible bounds for disease burden estimates [50,51,60].

Bayesian frameworks were used frequently to propagate uncertainty through posterior distributions of parameters and model structures, enabling the derivation of interval estimates and the exploration of uncertainties related to relationships between variables [44]. Bootstrapping methods were also commonly used, often applied to resampling procedures for generating confidence intervals at pixel or population levels and evaluating the robustness of predictions [45,54,61,75]. For example, ensemble modeling, including BRT and combinations of multiple machine learning models, accounted for uncertainty by training separate models on bootstrap datasets and combining outputs from top-performing models [61]. Cross-validation techniques were used to evaluate error rates and to avoid overfitting [33]. Additionally, uncertainty in intermediate measures such as endemicity, urbanization, and epidemic deaths was often incorporated through iterative model realizations in fully Bayesian frameworks, involving thousands of models runs to refine estimates [37,39,51].

## Stated Limitations

The studies identified several critical limitations in estimating disease burden, primarily related to data quality, methodological assumptions, and model uncertainties. Data is the first and most important limitation described. Incomplete or sparse data, particularly in resource-limited settings, lead to underreporting of cases and deaths [17,32,49,74]. Subclinical or mild cases were often missing in surveillance datasets, while the limited standardized diagnostic methods used in serological surveys lead to misclassification or lack specificity and restrict the accuracy of key metrics like prevalence and FOI [12,34,39,45,50,53,61]. Non-representative, outdated, or geographically restricted data further limited the precision of estimates, while variability in reporting rates across regions and healthcare systems introduced biases [49,52]. Additionally, the extrapolation of data from specific regions or populations to broader areas often ignored regional heterogeneity in healthcare access, disease awareness, and transmission dynamics [17,34].

Methodological limitations were also prominent, with models often relying on oversimplified assumptions about disease dynamics, uniform impacts of interventions, or constant transmission intensity [12,38,39,41,47,53,56,72]. Adjustment factors, such as multipliers or expansion factors, were based on limited empirical studies and varied widely across regions, populations, and disease severity [16,50,72]. Uncertainty in parameter estimates, model structure, and covariates, such as climatic or demographic factors, was not always fully propagated, resulting in wide confidence intervals or imprecise predictions [18,46,63]. Some studies also mentioned the focus on acute disease phases excluded long-term impacts like chronic symptoms or psychological consequences [34,49,71].

## Discussion

This scoping review provides a comprehensive overview of the current methodologies used to estimate the burden of acute febrile infectious diseases. The results highlight the diverse approaches employed across different diseases and geographic regions, ranging from traditional surveillance-based estimates to advanced modelling techniques incorporating multiple data sources. Malaria and dengue remain the most studied diseases, benefiting from relatively rich data availability and well-established estimation frameworks, while other febrile diseases such as leptospirosis, melioidosis and other more neglected diseases face significant challenges due to sparse data and limited surveillance capacities.

Burden estimates play a crucial role in guiding public health interventions, resource allocation, and research prioritization despite inherent limitations and uncertainties. While these estimates may not fully capture the complexity of disease transmission and underreporting, they provide a foundational understanding of the scale and impact of diseases, enabling comparisons across different diseases, regions, and time periods. They serve as valuable role in helping effective policy making and resource allocation. It is therefore essential to acknowledge the limitations and uncertainties while recognizing the importance of continued efforts to refine estimation methods and improve data quality. The increasing integration of environmental and vector ecology covariates reflects a broader methodological shift in burden estimation from adjustment-based approaches toward predictive geospatial modelling. While earlier methods primarily corrected surveillance data using expansion factors or demographic adjustments, contemporary frameworks leverage high-resolution environmental and climatic data to enable inference in regions lacking empirical observations. This transition has expanded the spatial granularity of burden estimates but also increased reliance on model assumptions and covariate quality.

This review addresses a critical gap in the literature where previous studies have largely focused on individual diseases such as malaria [15,100], dengue [19], and leishmaniasis [74]. By consolidating insights across multiple diseases, it offers a broader perspective on the methodological landscape, facilitating comparisons and identifying common challenges and best practices. A recurring methodological concern identified across the included studies is the reliance on extrapolation in data-scarce settings. In regions with limited surveillance infrastructure or sparse empirical observations, burden estimates are often generated by applying statistical relationships derived from data-rich contexts to areas lacking primary data. While such extrapolation is methodologically unavoidable in many global modelling exercises, it can introduce structural bias when underlying assumptions, such as homogeneous transmission dynamics, constant case-fatality ratios, or uniform healthcare access, do not reflect local realities. Acute tropical infectious diseases are highly sensitive to ecological, socio-economic, and health system variability, and oversimplified assumptions may obscure subnational heterogeneity or misrepresent disease burden in marginalized populations. Although global modelling frameworks have substantially advanced comparative burden estimation, the increasing reliance on cross-regional extrapolation highlights the importance of strengthening locally grounded data systems and incorporating context-specific parameters wherever possible. Without such contextual calibration, even technically sophisticated models risk reinforcing uncertainty rather than resolving it.

While malaria accounted for a large proportion of incidence and mortality modelling studies, comparatively few included studies estimated malaria DALYs using novel methodological frameworks. This likely reflects the dominance of standardized DALY calculations conducted within the GBD and WHO frameworks, which were excluded from this review when applied without additional methodological innovation. In contrast, dengue and chikungunya studies more frequently incorporated expansion factors, seroprevalence-based incidence estimation, or hybrid modelling approaches prior to DALY calculation, resulting in greater methodological heterogeneity and inclusion in this review.

Although formal risk-of-bias appraisal was not conducted, substantial variation in methodological robustness and transparency was observed across included studies. Earlier multiplier- and expansion factor–based approaches often relied on deterministic adjustment factors and scenario-based sensitivity analyses, with limited probabilistic uncertainty propagation. In contrast, more recent geospatial and transmission modelling studies frequently implemented Bayesian hierarchical frameworks, Markov Chain Monte Carlo simulations, ensemble modelling, and spatial cross-validation procedures.

These approaches typically reported posterior credible intervals or bootstrapped uncertainty bounds and explicitly propagated uncertainty across multiple modelling stages. External validation practices also varied: while ensemble and machine-learning models commonly incorporated cross-validation or bootstrap resampling, multiplier-based methods rarely included formal validation against independent datasets. The increasing adoption of probabilistic modelling frameworks reflects methodological maturation over time; however, heterogeneity in uncertainty quantification and assumption transparency should be considered when comparing burden estimates across studies.

Unlike disease-specific reviews, which often provide in-depth discussions on a single pathogen, our study integrates approaches used across diverse epidemiological contexts, highlighting variations in data sources, modeling techniques, and spatial scales. This synthesis allows for the identification of underutilized methodologies and opportunities for cross-disease learning, which can ultimately enhance the accuracy and applicability of burden estimates across different settings. Our review underscores the evolving nature of burden estimation methodologies, documenting the transition from traditional surveillance-based approaches to more sophisticated frameworks that leverage machine learning, Bayesian geostatistical models, and integrating data from multiple sources.

### Limitation

This scoping review has several limitations that should be considered when interpreting the findings. First, the search strategy was restricted to PubMed and to English-language, peer-reviewed publications. Although PubMed captures a large proportion of biomedical and infectious disease research, relevant modelling studies indexed in other databases as well as interdisciplinary environmental and geospatial modelling journals, may have been missed. In addition, the exclusion of non-English publications may have led to underrepresentation of locally developed burden estimation approaches, particularly from regions where acute tropical infectious diseases are highly endemic. Second, this review intentionally excluded studies that directly applied standardized WHO or GBD DALY calculation methods without additional methodological innovation. While this criterion allowed us to focus on methodological development and novel analytical frameworks, it may underrepresent the broader landscape of routine burden estimation practice, particularly for diseases such as malaria where centralized GBD modelling dominates. Third, consistent with scoping review methodology, we did not perform a formal quality appraisal or risk-of-bias assessment of included studies. As a result, while we describe methodological approaches and trends, we do not evaluate the comparative robustness, predictive performance, or validity of individual modelling frameworks. Fourth, the distribution of included studies was heavily skewed toward malaria and dengue, which together accounted for the majority of identified literature. As a result, the methodological patterns described in this review, particularly the prominence of geospatial and Bayesian modelling frameworks, largely reflect approaches developed for these high-burden, vector-borne diseases. For other acute tropical infectious diseases, where empirical data and modelling efforts remain sparse, methodological diversity may be underrepresented. This imbalance limits the generalizability of our conclusions across all acute tropical infectious diseases and reflects broader research and funding disparities that shape the global modelling landscape. Fifth, the initial screening process and extraction were conducted primarily by one reviewer, with consultation for uncertain cases. Although predefined inclusion and exclusion criteria were applied to minimize subjectivity, some degree of screening bias cannot be fully excluded.

The field of infectious disease modelling is rapidly evolving, particularly with the increasing application of machine learning, artificial intelligence, and real-time data platforms. As such, this review represents a snapshot of the methodological landscape up to the most recent search date, and emerging approaches may not yet be fully captured in the published literature. An additional class of burden estimation approaches not prominently represented in this scoping review are time series–based excess mortality methods, which have been widely applied to estimate mortality attributable to influenza and other respiratory infections [101]. These approaches use long-term mortality time series, often combined with sentinel surveillance or pathogen activity indicators, to estimate attributable excess deaths above an expected baseline during epidemic periods. Where high-quality civil registration systems and consistent surveillance data exist, time series methods

offer important advantages, including reduced reliance on individual-level cause-of-death attribution and the ability to capture indirect or misclassified deaths. However, their applicability to many acute febrile infectious diseases remains limited, particularly in low- and middle-income country settings, where long-term mortality time series, pathogen-specific surveillance, or reliable death registration systems are sparse. As surveillance systems for febrile illnesses expand, time series–based approaches may nonetheless represent a valuable complement to existing burden estimation frameworks.

## Recommendations for Future Research

To advance the estimation of acute febrile disease burden, several critical factors should be considered, and we offer a set of recommendations (Table 3).

The first and most fundamental step is data collection and management, as the quality and comprehensiveness of available data directly impact the validity of the estimates. A critical starting point is a comprehensive baseline field assessment for informing appropriate data collection plans. A thorough review, understanding and assessing the existing data landscape is necessary including routine surveillance systems, hospital, and health facility records, academic literature, public reporting, and survey results, etc. Particular attention should be given to identifying and addressing key geographic data gaps, notably in regions such as Central Africa and Central Asia, where surveillance infrastructure is limited or absent. Before initiating burden estimation efforts, it is important to evaluate the data quality, representativeness, and potential biases. For example, understanding the nature of data collection, whether passive or active surveillance, and clarifying definitions, reporting standards, and data management practices are crucial for ensuring data reliability and comparability. And surveillance systems often suffer from inherent biases, such as underreporting and inconsistencies in data collection across different regions, population groups and settings. In regions lacking reliable data, it may be necessary to carry out prospective studies and surveys to fill critical gaps and collect empirical data. In settings with limited resources,

**Table 3. Key recommendations for advancing acute tropical infectious disease burden estimation.**

| Recommendation Area | Priority | Summary of Recommendations | Specific Action for Acute Tropical NTDs |
|---|---|---|---|
| Data Collection | Immediate | Baseline field assessment; quality evaluation | Conduct baseline field assessments in data-sparse endemic regions (e.g., Central Africa, parts of Southeast Asia) to reduce reliance on cross-regional extrapolation. |
| Surveillance | Immediate | Strengthen active/passive systems; sentinel sites | Strengthen sentinel surveillance and laboratory-confirmed case reporting for acute febrile illnesses to improve model calibration inputs. |
| Metrics Selection | Immediate | Standardize burden metrics (incidence, DALYs, mortality) | Standardize reporting of incidence, mortality, and DALYs with explicit uncertainty intervals to enable cross-disease comparability. |
| Uncertainty Quantification | Immediate | Sensitivity analysis; cross-validation | Require probabilistic uncertainty propagation (e.g., Monte Carlo or Bayesian posterior intervals) in future burden estimation studies. |
| Data Sharing and Accessibility | Medium-term | International data-sharing; open-access repositories | Establish regionally coordinated data platforms with local co-ownership and standardized reporting protocols to improve transparency and reproducibility. |
| Spatial Resolution | Medium-term | Align resolution with data availability | Align modelling resolution with surveillance capacity, avoiding over-interpretation of high-resolution outputs in data-poor settings. |
| Covariate Integration | Medium-term | Combine clinical, environmental, policy covariates | Prioritize environmental and vector ecology covariates for vector-borne NTDs while incorporating healthcare-access adjustments for underreporting correction. |
| Hybrid Modeling | Medium-term | Integrate traditional and novel data sources | Combine surveillance correction methods (expansion factors) with spatial predictive modelling to reduce bias in under-ascertained diseases. |
| Advanced Analytics | Long-term | Apply machine learning & AI for prediction | Develop shared modelling platforms that allow local institutions to input contextual parameters into validated predictive frameworks. |
| Framework Standardization | Long-term | Adaptable estimation frameworks | Develop adaptable, disease-agnostic modelling templates for acute tropical infections to enhance comparability across settings. |
| Collaborative Efforts | Structural | Interdisciplinary researcher-policymaker partnerships | Promote equitable partnerships between local institutions and international modelling groups to support sustainable analytical capacity. |

sentinel surveillance systems based on well-selected, demographically representative sites may serve as a cost-effective approach for tracking long-term disease trends.

Improving data sharing and accessibility is another critical aspect of enhancing burden estimation efforts. Establishing international data-sharing networks can help harmonize data collection efforts and promote collaboration across regions. Given financial and logistical constraints, an alternative solution is the systematic publication of datasets in open-access repositories, enabling broader utilization by researchers and policymakers. Existing initiatives that collate publicly available disease datasets, such as OpenDengue.org [102], which provides an openly accessible global dengue surveillance database; the Infectious Diseases Data Observatory (IDDO) [103], which curates and standardizes individual-level infectious disease data; large-scale population survey platforms such as DHS and MICS; and the WHO Global Health Observatory, which compiles country-level health indicators, demonstrate the feasibility and value of such efforts and offer a model for similar platforms for other acute infectious diseases.

Once a robust data foundation is established, selecting appropriate methodologies is essential for generating reliable and actionable burden estimates. Choosing the right burden metrics, such as incidence, mortality, or DALYs, is crucial for enabling cross-country and cross-disease comparisons. DALYs, introduced during the first Global Burden of Disease (GBD) study in 1992 [86,104], provide a comprehensive measure that accounts for both premature mortality and years lived with disability, making them a valuable tool for health prioritization. In addition to selecting the appropriate metric, determining the spatial resolution and extent of burden estimation should be based on data availability and the availability of covariates. High-resolution data at pixel or point levels can provide detailed insights for diseases with robust surveillance systems, while coarser resolutions, such as country or regional levels, may be necessary in data-limited settings.

Comprehensive consideration of relevant covariates in essential reflecting the multifaceted nature of disease burden, spanning clinical, epidemiological, environmental, and policy-related aspects. Incorporating diagnostic accuracy (sensitivity and specificity), healthcare access, and the availability of interventions such as vaccines and vector control measures allows for better adjustment of estimates and further utility for guiding control efforts. Hybrid modelling approaches that integrate both traditional data sources and novel elements, such as environmental monitoring, and mobility data, can enhance model adaptability across different epidemiological settings. As new technologies and interventions continue to evolve, burden estimation should also consider interdisciplinary collaboration and incorporating advanced analytics, such as artificial intelligence and machine learning, which can significantly enhance the capacity to analyse complex interactions between factors, help refine predictions and identify hidden patterns in disease transmission. Moreover, uncertainty quantification should be a component of methodological frameworks and result reporting, ensuring that estimates account for variability in data quality, model assumptions, and parameter inputs and keep clearly reported to aid in informed decision-making.

Beyond methodological refinement, strengthening local institutional capacity is essential for sustainable and context-sensitive burden estimation. Many acute tropical infectious diseases disproportionately affect regions with limited modelling infrastructure, yet analytical leadership often remains concentrated in resource-rich institutions. While advanced modelling frameworks can compensate for sparse data, reliance on externally driven modelling without local co-development may limit contextual interpretation and long-term sustainability. Collaborative platforms that allow local institutions to input region-specific parameters, adapt assumptions to local epidemiology, and retain ownership of data inputs and analytical outputs are therefore critical. Capacity-building initiatives and shared modelling infrastructures can help bridge technical gaps while mitigating ethical concerns related to extractive data practices. Such partnerships move burden estimation beyond purely technical exercise toward a more equitable and locally grounded public health tool.

To summarise, in the short term, efforts should focus on integrating multiple data sources, such as surveillance records, environmental data, and academic literature, while applying a range of modelling approaches to identify the most suitable methods for different contexts. Iterative refinement through cross-validation and sensitivity analyses can help improve estimate robustness. In the long term, establishing standardized frameworks for burden estimation that are adaptable across

diseases and regions will facilitate comparability and consistency. Moreover, investing in better surveillance infrastructure and capacity-building is essential, especially in countries currently lacking adequate or fit-for-purpose surveillance systems. Collaboration among researchers, policymakers, and data providers is crucial to ensure that burden estimates are not only accurate but also actionable for health planning and resource allocation. Encouraging transparent data sharing and methodological harmonization through interdisciplinary partnerships can enhance the reliability and usability of burden estimates, ultimately supporting more effective disease control and prevention strategies.

## Conclusion

This scoping review systematically mapped methodological approaches used to estimate the burden of acute tropical infectious diseases, identifying 60 eligible studies spanning diverse geographic regions and modelling frameworks. The literature was heavily concentrated on malaria and dengue, reflecting both disease burden and research investment disparities. Across studies, we observed a clear methodological progression from surveillance-based multiplier and expansion factor approaches toward increasingly sophisticated Bayesian geospatial, ensemble, and machine-learning models with formal uncertainty propagation. Covariate integration evolved accordingly, with earlier models emphasizing demographic and healthcare-access adjustments, and more recent frameworks incorporating environmental and vector ecology variables to enable spatial prediction in data-sparse settings. Despite methodological advances, substantial heterogeneity persists in uncertainty quantification, model validation, and transparency of assumptions. Strengthening locally grounded data systems, equitable modelling partnerships, and standardized reporting of uncertainty will be critical for improving the robustness and interpretability of burden estimates for underrepresented acute tropical infectious diseases.

## Supporting information

**S1 Table. Preferred reporting items for systematic reviews and meta-analyses extension for scoping reviews (PRISMA-ScR) checklist.**
(DOCX)

**S2 Table. Full search strategy.**
(DOCX)

**S3 Table. Comprehensive list of studies included in the review, with ID corresponding to citation number in the main reference list (n = 60).**
(DOCX)

**S4 Table. Key information extracted from included studies, with ID corresponding to citation number in the main reference list (n = 60).**
(DOCX)

**S5 Table. List of articles excluded at full-text review stage and corresponding justification based on predefined eligibility criteria.**
(DOCX)

**S6 Table. General characteristics of included studies (n = 60).**
(DOCX)

**S1 Fig. Distribution of modelling approach categories across three time periods (Pre-2010, 2010–2017, 2018–2025) among included studies.**
(DOCX)

## Acknowledgments

We would like to thank Elinor Harriss from the Bodleian Health Care Libraries of University of Oxford for her assistance with development of the search strategy and review of the protocol.

## Author contributions

**Conceptualization:** Richard James Maude, Nicholas Philip John Day, Benn Sartorius.

**Data curation:** Qian Wang.

**Formal analysis:** Qian Wang.

**Funding acquisition:** Qian Wang, Richard James Maude, Nicholas Philip John Day, Benn Sartorius.

**Investigation:** Qian Wang.

**Methodology:** Qian Wang, Richard James Maude, Benn Sartorius.

**Project administration:** Richard James Maude, Nicholas Philip John Day.

**Supervision:** Richard James Maude, Nicholas Philip John Day, Benn Sartorius.

**Visualization:** Qian Wang.

**Writing – original draft:** Qian Wang.

**Writing – review & editing:** Qian Wang, Richard James Maude, Nicholas Philip John Day, Benn Sartorius.

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
