## [Decision Letter · Decision Letter 0]

3 Oct 2025

PNTD-D-25-01169

Methods for estimating the burden of acute infectious diseases: a scoping review

Dear Dr. Wang,

Thank you for submitting your manuscript to PLOS Neglected Tropical Diseases. After careful consideration, we feel that it has merit but does not fully meet PLOS Neglected Tropical Diseases's publication criteria as it currently stands. Therefore, we invite you to submit a revised version of the manuscript that addresses the points raised during the review process.

Please submit your revised manuscript within 60 days Dec 02 2025 11:59PM. If you will need more time than this to complete your revisions, please reply to this message or contact the journal office at plosntds@plos.org. Please include the following items when submitting your revised manuscript:

We look forward to receiving your revised manuscript.

Kind regards,

Katie Anders

Academic Editor

Qu Cheng

Section Editor

Shaden Kamhawi

co-Editor-in-Chief

Paul Brindley

co-Editor-in-Chief

**Additional Editor Comments:**

In preparing your revised manuscript for resubmission, please address in full the following comments in addition to the comments from the two reviewers:

1. As raised by both reviewers, and required in item 8 of the PRISMA-ScR Checklist, please ensure you provide in your supplementary material the full electronic search strategy, such that it could be repeated.

2. As noted by reviewer 2, the current manuscript title doesn't accurately reflect the scope of diseases considered in your scoping review. Please adjust the title accordingly, e.g. "Methods for estimating the burden of acute *tropical* infectious diseases: a scoping review", and adjust the corresponding wording throughout the manuscript.

3. A key paper is missing from the included studies, which was the first study to apply a formal modelling framework to map the global distribution of dengue disease burden: Bhatt S, Gething PW, Brady OJ et al (2013) Nature. https://doi.org/10.1038/nature12060, which raises a concern about the search strategy and/or screening and assessment process. Please investigate why this paper was not identified and/or included in the scoping review, and whether this indicates any systematic issues with the review methodology.

4. The Global Burden of Disease methodology and the WHO DALYs calculation methodology should be introduced briefly in the introduction, before the subsequent reference in the Methods section to the exclusion of studies that directly applied these methodologies without any additional modelling or methodological advancement (line 118-122)

5. Methods Line 103: "The original search was done in 30, March 2024." The word 'original' implies that a second or updated search may have been done later - please provide details of any subsequent search if so, or adjust wording to remove this ambiguity if not. Also, the methods state publications up to 2024 (presumably March) were included, but in results section no publications after 2022 (Fig 2; line 166). If this is correct, please state in results that there were no eligible publications found in 2023-24.

6. Methods Line 125: "(3) the article didn't measure any health burden (e.g., economic cost or social burden)" - this wording is confusing, does it mean to say "e.g. only measured economic cost or social burden"? Please improve the wording.

7. Figure 1: the last box in the Screening section of the flowchart ('Additional records identified through articles' references') seems like it belongs up in the Identification section, as a secondary means of identifying records that (presumably) then would have been screened and assessed for eligibility in the same way as the 1756 records identified from PubMed. The current flowchart implies that these records were directly included in the review without screening/eligibility assessment.

8. Results Line 157: "A total of 1,756 records were identified via the search strategies of PubMed and loaded into Rayyan." Presuming that Rayyan is a software tool, this detail should be provided in the methods, not in the results, and with an explanation of what this software was used for.

9. The use of the term 'prevalence' throughout the manuscript conflates different measures: please be specific in referring to e.g. 'parasite prevalence' for malaria (e.g. line 195), 'dengue seroprevalence' where that is what you mean (e.g. line 208).

10. Results Line 196-7: the description of the diverse base data sources used in malaria burden estimation states that this includes "...modelled maps of disease risk (19-36). These data are sourced from a combination of large-scale sources like the Malaria Atlas Project (MAP) (31,36)....". The 'modelled maps of disease risk' produced by MAP and others are not base data, rather they are the modelled burden estimation outputs that are the subject of this scoping review. References #31 and 36 cited here report geospatial modelling estimates of malaria mortality and incidence, respectively. Similarly at line 218 in the discussion of base data sources for dengue burden estimation, "....fine-grained grid-based resolutions (e.g., 5x5 km grids) (49-51)...", these are model-based outputs at 5x5 km pixel resolution, not base data. Improved clarity is needed in the discussion of base data INPUTS (i.e. geolocated data on disease occurrence, parasite prevalence, incidence) versus the modelled maps of disease risk that are the OUTPUTS of disease burden modelling.

11. Table 2. It seems surprising that there were no studies included with estimates of DALYs attributable to malaria. Presumably this is because all such studies met the criteria for exclusion due to direct application of the WHO DALYs calculation method without any modification - whereas more diverse methods were applied for dengue DALY estimation? This warrants mention in the results or discussion section.

12. As raised by the reviewers, the Covariates section of the results (including Table 3) is overly descriptive and lacks analytical synthesis. Please make clearer how the consideration of covariates relates to the evolving methodologies for disease burden estimation; i.e. environmental and vector ecology covariates are a critical input for geospatial modeling and machine learning methods, as they inform inference of disease burden in areas without empirical data based on observations in other locations. Whereas other/older approaches based on expansion factors etc may have considered only sociodemographic and/or health system characteristics or even no covariates at all.

13. Items 17 (Results of individual sources of evidence) and 20 (Limitations) in the PRISMA-ScR checklist (Supplementary material) have been marked as N/A, which is not an adequate response. Please address these items in your revised manuscript. Item 17 requires that "For each included source of evidence, present the relevant data that were charted that relate to the review questions and objectives". This should be addressed by including in Supplementary Table S2 the other relevant characteristics that were charted for each study, as described in your methods (lines 130-5), specifically the method for disease burden estimation, base data sources, covariates, and techniques to address uncertainty, for each record. Please also make consistent the format of the 'Authors' column in Table S2, and add the citation number corresponding to the reference list in the main manuscript.

14. The discussion of the need for improved data sharing and accessibility at lines 432-6 would be strengthened by mentioning existing initiatives to collate publicly available datasets, such as opendengue.org (https://doi.org/10.1038/s41597-024-03120-7)

**Journal Requirements:**

At this stage, the following Authors/Authors require contributions: Qian Wang, Richard James Maude, Nicholas Philip John Day, and Benn Sartorius. Please ensure that the full contributions of each author are acknowledged in the "Add/Edit/Remove Authors" section of our submission form.

3) Please ensure that the funders and grant numbers match between the Financial Disclosure field and the Funding Information tab in your submission form. Note that the funders must be provided in the same order in both places as well.

State the initials, alongside each funding source, of each author to receive each grant. For example: "This work was supported by the National Institutes of Health (####### to AM; ###### to CJ) and the National Science Foundation (###### to AM).".

**Reviewers' Comments:**

Reviewer's Responses to Questions

**Key Review Criteria Required for Acceptance?**

**Methods**

-Are the objectives of the study clearly articulated with a clear testable hypothesis stated?

-Is the study design appropriate to address the stated objectives?

-Is the population clearly described and appropriate for the hypothesis being tested?

-Is the sample size sufficient to ensure adequate power to address the hypothesis being tested?

-Were correct statistical analysis used to support conclusions?

-Are there concerns about ethical or regulatory requirements being met?

Reviewer #1: (No Response)

Reviewer #2: Please, see "Summary and General Comments"

**Results**

-Does the analysis presented match the analysis plan?

-Are the results clearly and completely presented?

-Are the figures (Tables, Images) of sufficient quality for clarity?

Reviewer #1: (No Response)

Reviewer #2: Please, see "Summary and General Comments"

**Conclusions**

-Are the conclusions supported by the data presented?

-Are the limitations of analysis clearly described?

-Do the authors discuss how these data can be helpful to advance our understanding of the topic under study?

-Is public health relevance addressed?

Reviewer #1: (No Response)

Reviewer #2: Please, see "Summary and General Comments"

**Editorial and Data Presentation Modifications?**

Reviewer #1: (No Response)

Reviewer #2: (No Response)

**Summary and General Comments**

Reviewer #1: Introduction

1. A few sentences providing background on the existing burden of two major NTDs, malaria and dengue, in endemic regions, and highlighting how gaps in precise burden estimates have slowed global control and elimination efforts, would strengthen the rationale for this study.

Methods

2. The authors should justify why PubMed was considered an exhaustive source for this review. While PubMed indexes a wide range of biomedical research, it does not capture all studies relevant to the research question. For instance, journals such as Environmental Research Letters publish work on environmental determinants of health outcomes, including infectious diseases, but not all such articles are indexed in PubMed. Standard practice is to include at least one additional database such as Scopus or Web of Science to complement PubMed search outputs. Please explain why other databases were excluded.

3. Why was the initial screening conducted by only one researcher? This approach may introduce bias. Both JBI and Cochrane recommend at least two researchers for initial screening, with an additional reviewer involved for conflict resolution when necessary.

4. The authors state that the study design was selected to address specific research questions. However, explicit reference to a standard scoping review framework, such as the Population, Concept, and Context (PCC) framework, would strengthen the robustness of the study design.

5. Please include the full search strategy in the supplementary material. This would enhance transparency and support future researchers in replicating or extending the study.

6. The manuscript notes that a biomedical information specialist (librarian) contributed to protocol development. However, I did not find a reference to the protocol itself. If I have overlooked it, please clarify; otherwise, it is recommended that the review protocol be made publicly available and cited to strengthen the rigor of the work.

Results

7. If possible, reconsider the color palette in Figure 2. Specifically, the colors used to represent dengue and malaria, as well as JE and Q fever, are difficult to distinguish at a glance. If possible, a network diagram (for example, a Sankey diagram) would be a useful way to summarize the results. The diagram could show how study regions link to the NTDs examined, then to the models or methods used, and finally branch out to the parameters included in the models or outcomes reported (such as mortality, incidence, or DALYs).

8. While the main paper includes a section to summarise the models used for burden estimation, supplementary materials should also include a column specifying the model or method used for estimation in each selected study. At present, this information is missing.

Discussion

9. In the results section, the authors note issues with extrapolation in data-scarce regions, often based on simplistic assumptions that ignore local context. This is particularly relevant to the GBD strategy from IHME, which has been criticized for missing contextual nuances and sometimes producing inaccurate results. This issue could be elaborated further, especially in the paragraph preceding the recommendations section. Highlighting the fallacy of overly simplistic assumptions and the overuse of extrapolation, rather than marking use of available sources to generate primary estimates would enrich the discussion.

10. While this review successfully maps existing methods for estimating the burden of acute infections, the discussion could be more analytical rather than purely descriptive. The recommendations, in particular, could be framed more equitably by emphasizing partnerships with local policymakers and scientific experts. The authors identify data-related challenges and suggest improvements focused largely on data collection, management, and sharing. However, local institutional capacity development should also be considered. While it may be unrealistic to expect every institution in these settings to have expert modelers or epidemiologists, systems could be established where experts from resource-rich contexts support the development of platforms based on advanced modeling strategies. Local institutions could then input relevant parameters and receive burden estimates in a clear, interpretable form. Such an approach could mitigate ethical concerns around external data use/sharing and foster equitable partnerships between local institutions and supporting collaborators.

Reviewer #2: This scoping review delivers a timely and comprehensive map of how acute NTD burden has been estimated over the past three decades. Its strengths lie in the systematic collation of methodological trends, the clear geographic and disease distributions, and the transparent reporting of covariates and uncertainty techniques. Below are some topic-by-topic suggestions for improvement.

Introduction

The title suggest that the scoping review covers acute infectious diseases broadly. However, the inclusion criteria in the Methods section specify that only acute infectious diseases listed in the NTDs recognized by PLOS Neglected Tropical Disease are considered. This limitation must be clearly stated in the Introduction or Methods to avoid misleading readers into thinking the review covers all acute infectious diseases (e.g., COVID-19, influenza, Ebola).

Method

1.Search Strategy Insufficiently Comprehensive： The search was limited to PubMed, potentially omitting relevant studies indexed in other major databases such as Web of Science, Embase, or Scopus. Additionally, there is no mention of whether grey literature (e.g., government reports) was searched, which is particularly important for capturing unpublished or locally conducted research.

2.The manuscript states that search strings covered "outcomes," "methods," and "diseases," but the actual search terms or a full search query (e.g., for PubMed) are not provided. Including the full search strategy as supplementary material is essential for transparency and reproducibility.

3.Incomplete Reporting of Screening Process: While the manuscript includes a PRISMA flowchart and numbers at each stage, it does not describe quality control measures during data extraction, such as whether dual independent screening and extraction were conducted.

4.Although scoping reviews do not require formal quality assessment, a brief descriptive summary or commentary on the methodological quality of included studies would help readers interpret the strength of the evidence.

Results

1.The manuscript included a study from 1987 (B.M. Greenwood et.al.) in Table S2, but Table 1 (Publication year distribution) only covers studies from 1998 onwards. Furthermore, the Methods section states that the search was conducted in PubMed for publications from 1990 to 2024. Please clarify how the 1987 study was identified and included.

2.Figure 2 and Table 1 appear to contain overlapping information regarding the distribution of included studies by publication year. We recommend integrating this information into a single, clear visual or moving one of them to the supplementary materials to avoid repetition and enhance the flow of the results section.

3.The section on covariates is a useful list but lacks analysis. A brief discussion on how the selection of covariates is rationalized for different diseases and settings, and the relative importance of different covariate types (environmental, demographic, healthcare-related).

4.The manuscript claims that burden-estimation methods have shifted from traditional surveillance to advanced approaches such as machine-learning and Bayesian geospatial models, yet this evolution is described only qualitatively. No quantitative evidence—e.g., counts or proportions of studies using each method type by time period, a stacked bar chart, or trend test—is provided.

Discussion

1.Disease coverage was highly skewed: malaria and dengue accounted for 76 % of all included studies, whereas other NTDs (e.g., leptospirosis, Melioidosis) were represented by only a handful of cases. This imbalance markedly limits the generalisability of the findings, yet the manuscript does not discuss how it may have shaped the overall conclusions.

2.Table 4 repeats well-known imperatives such as "Data sharing and accessibility" and "Apply machine learning". Indicate which of these actions are new or prioritised for acute NTDs. I recommend strengthening specifying actionable public health measures.

3.As a scoping review, the inclusion criteria—limited to English-language, peer-reviewed articles—may introduce language and publication biases. Relevant non-English regional studies or grey literature containing unique estimation methods might have been excluded. Furthermore, restricting the search to PubMed alone, without including other major databases such as Embase or Scopus, could lead to the omission of significant literature. It is recommended to address these potential "literature selection biases" in the Limitations section and discuss their possible impact on the representativeness of the conclusions.

Conclusion

While the Conclusion provides a general summary of the study’s contributions and future directions, it would benefit from a more explicit integration of key results presented in the manuscript.

PLOS authors have the option to publish the peer review history of their article (what does this mean?). If published, this will include your full peer review and any attached files.

Reviewer #1: **Yes:** Dinesh Bhandari

Reviewer #2: No

**Figure resubmission:**
---

## [Editor Report · Decision Letter 1]

5 Apr 2026

PNTD-D-25-01169R1
Methods for estimating the burden of acute tropical infectious diseases: a scoping review
PLOS Neglected Tropical Diseases
 
Dear Dr. Wang,

Thank you for submitting your manuscript to PLOS Neglected Tropical Diseases. After careful consideration, we feel that it is suitable for publication in PLOS Neglected Tropical Diseases after a few additional points are addressed, as detailed below.

* A letter that responds to each point raised by the editor and reviewer(s). You should upload this letter as a separate file labeled 'Response to Reviewers'. This file does not need to include responses to any formatting updates and technical items listed in the 'Journal Requirements' section below.* A marked-up copy of your manuscript that highlights changes made to the original version. You should upload this as a separate file labeled 'Revised Manuscript with Track Changes'.* An unmarked version of your revised paper without tracked changes. You should upload this as a separate file labeled 'Manuscript'.If you would like to make changes to your financial disclosure, competing interests statement, or data availability statement, please make these updates within the submission form at the time of resubmission. Guidelines for resubmitting your figure files are available below the reviewer comments at the end of this letter.We look forward to receiving your revised manuscript.Kind regards,Katie AndersAcademic EditorPLOS Neglected Tropical DiseasesQu ChengSection EditorPLOS Neglected Tropical Diseases

Shaden Kamhawi

co-Editor-in-Chief

Paul Brindley

co-Editor-in-Chief

**Additional Editor Comments :**
Thank you for addressing the two reviewers' comments and additional editor's comments very thoroughly in your revised manuscript.

There are only a few minor additional points requiring your attention, in order for the manuscript to be accepted for publication:

1. Table S5: this is a valuable addition. However there is a switch halfway through the table from providing the full citation details and explicit reasons for exclusion, to instead providing only the paper title and a nondescript reason for exclusion. Please revise to include the full information for all excluded publications listed in Table S5

2. Line 130: "The original search was done in 30, March 2024" -> "The original search was done on 30 March 2024"

3. Table 3: update table title to align with revised manuscript title 'Table 3. Key Recommendations for Advancing Acute *Tropical* Infectious Disease Burden Estimation'

4. Regarding Reviewer 1's comment #10, one minor addition is suggested at line 646:

"Collaborative platforms that allow local institutions to input region-specific parameters, adapt assumptions to local epidemiology, and retain ownership of *data inputs and* analytical outputs are therefore critical."

5. It is recommended that you cite the new supporting information Figure S1 at an appropriate place in the manuscript text, though it is not a journal requirement that supporting information is cited.
**Journal Requirements:**

1) Please ensure that the uploaded supplementary files are without tracked changes or highlighting.

**Figure resubmission:**While revising your submission, we strongly recommend that you use PLOS’s NAAS tool (https://ngplosjournals.pagemajik.ai/artanalysis) to test your figure files. NAAS can convert your figure files to the TIFF file type and meet basic requirements (such as print size, resolution), or provide you with a report on issues that do not meet our requirements and that NAAS cannot fix.

After uploading your figures to PLOS’s NAAS tool - https://ngplosjournals.pagemajik.ai/artanalysis, NAAS will process the files provided and display the results in the "Uploaded Files" section of the page as the processing is complete. If the uploaded figures meet our requirements (or NAAS is able to fix the files to meet our requirements), the figure will be marked as "fixed" above. If NAAS is unable to fix the files, a red "failed" label will appear above. When NAAS has confirmed that the figure files meet our requirements, please download the file via the download option, and include these NAAS processed figure files when submitting your revised manuscript.**Reproducibility:**
To enhance the reproducibility of your results, we recommend that authors of applicable studies deposit laboratory protocols in protocols.io, where a protocol can be assigned its own identifier (DOI) such that it can be cited independently in the future. Additionally, PLOS ONE offers an option to publish peer-reviewed clinical study protocols. Read more information on sharing protocols at https://plos.org/protocols?utm_medium=editorial-email&utm_source=authorletters&utm_campaign=protocols

---

## [Editor Report · Decision Letter 2]

11 Apr 2026

Dear ms Wang,

We are pleased to inform you that your manuscript 'Methods for estimating the burden of acute tropical infectious diseases: a scoping review' has been provisionally accepted for publication in PLOS Neglected Tropical Diseases.

Best regards,

Katie Anders

Academic Editor

Qu Cheng

Section Editor

Shaden Kamhawi

co-Editor-in-Chief

Paul Brindley

co-Editor-in-Chief

---

## [Editor Report · Acceptance letter]

Dear ms Wang,

We are delighted to inform you that your manuscript, "Methods for estimating the burden of acute tropical infectious diseases: a scoping review," has been formally accepted for publication in PLOS Neglected Tropical Diseases.

Best regards,

Shaden Kamhawi

co-Editor-in-Chief

Paul Brindley

co-Editor-in-Chief
